# Nanoconfinement-triggered oligomerization pathway for efficient removal of phenolic pollutants via a Fenton-like reaction

Xiang Zhang[1,5], Jingjing Tang[1,5], Lingling Wang[1,5], Chuan Wang[1], Lei Chen[2], Xinqing Chen[3], Jieshu Qian[1,2,4] ✉ & Bingcai Pan[2] ✉

Heterogeneous Fenton reaction represents one of the most reliable technologies to ensure water safety, but is currently challenged by the sluggish Fe(III) reduction, excessive input of chemicals for organic mineralization, and undesirable carbon emission. Current endeavors to improve the catalytic performance of Fenton reaction are mostly focused on how to accelerate Fe(III) reduction, while the pollutant degradation step is habitually overlooked. Here, we report a nanoconfinement strategy by using graphene aerogel (GA) to support UiO-66-NH$_2$-(Zr) binding atomic Fe(III), which alters the carbon transfer route during phenol removal from kinetically favored ring-opening route to thermodynamically favored oligomerization route. GA nanoconfinement favors the Fe(III) reduction by enriching the reductive intermediates and allows much faster phenol removal than the unconfined analog (by 208 times in terms of first-order rate constant) and highly efficient removal of total organic carbon, i.e., 92.2 ± 3.7% versus 3.6 ± 0.3% in 60 min. Moreover, this oligomerization route reduces the oxidant consumption for phenol removal by more than 95% and carbon emission by 77.9%, compared to the mineralization route in homogeneous Fe$^{2+}$+H$_2$O$_2$ system. Our findings may upgrade the regulatory toolkit for Fenton reactions and provide an alternative carbon transfer route for the removal of aqueous pollutants.

Fenton reaction is of extensive interest for biomedicine[1], gene expression[2], sensing[3], material preparation[4], chemical synthesis[5], and environmental remediation[6]. In particular, the seminal version of Fenton reaction involving Fe$^{2+}$ and H$_2$O$_2$ has realized global application in decentralized water treatment systems to ensure water safety because the generated hydroxyl radicals (•OH) (oxidation potential: 2.80 V, half-life: <5 μs)[7,8] with strong oxidizing ability could degrade recalcitrant and toxic organic pollutants under ambient conditions[9].

However, this reaction suffers intrinsic drawbacks including non-recyclable catalyst, narrow pH suitability, and easy formation of Fe sludge that breaks the redox cycle[10]. Promising alternatives include heterogeneous Fenton processes that allow facile recycle of catalyst and extension of pH suitability[11]. Nevertheless, two fundamental issues exist in heterogeneous Fenton reactions. The first is the sluggish Fe(III) reduction reaction, which is the rate-limiting step in H$_2$O$_2$ activation[12]. Currently, extensive efforts have been devoted to accelerating the

[1]Jiangsu Key Laboratory of Chemical Pollution Control and Resources Reuse, School of Environmental and Biological Engineering, Nanjing University of Science and Technology, Nanjing 210094, China. [2]Research Center for Environmental Nanotechnology (ReCENT), State Key Laboratory of Pollution Control and Resources Reuse, School of Environment, Nanjing University, Nanjing 210023, China. [3]CAS key Laboratory of Low-carbon Conversion Science and Engineering, Shanghai Advanced Research Institute, Chinese Academy of Sciences, Shanghai 201210, China. [4]School of Environmental Engineering, Wuxi University, Jiangsu 214105, P. R. China. [5]These authors contributed equally: Xiang Zhang, Jingjing Tang, Lingling Wang. ✉e-mail: qianjieshu@foxmail.com; bcpan@nju.edu.cn

Fe(III) reduction through either providing external chemical reagents[13] and/or light/electrical energy[7,14], or designing high-efficiency composite catalysts[15,16]. Another issue is the excessive input of chemicals, i.e., catalyst and $H_2O_2$, to achieve complete mineralization of aqueous pollutants with unregulated carbon emission. Attempts to address this issue include the modulation of catalytic active sites to improve the $H_2O_2$ activation efficiency[17], and the change of $H_2O_2$ activation pathways for a higher pollutant selectivity[18,19]. These endeavors are normally focused on one aspect of the reaction, i.e., $H_2O_2$ activation, for nominally improved pollutant degradation rate, while the other side of the reaction, i.e., pollutant degradation, is habitually overlooked.

The recent flourishment in nanoscience and nanotechnology has stimulated scientists to create various nanoconfined systems to mimic natural biological systems to achieve extraordinary rate acceleration and specificity for efficient chemistry[20,21]. Nanoconfinement fundamentally changes the chemical and physical properties of the confined molecules[22], and more importantly, poses profound influences on the confined chemical reactions from both kinetics and thermodynamic aspects[23,24]. Chemical reactions could be accelerated in nanoconfinement due to the increase of local concentration of the reactants[25,26]. The restriction of the nanoconfinement could limit the size and the shape of the resulting products, enhancing or altering the product selectivity[27–29]. Furthermore, nanoconfinement could affect reaction pathways by stabilizing reactive intermediates, achieving rate acceleration and enhanced selectivity[30–32]. For examples, the nanoconfinement by open Cu nanocavity could stabilize C2 intermediates in the CO electroreduction process, promoting C3 alcohol formation[31]. The nanoconfinement of a carbon layer surrounding a $CuO_x$ catalyst could dictate the key intermediate *HOCCH via the hydrogenation pathway, thus achieving high ethanol selectivity in the $CO_2$ electroreduction reaction[32]. Based on the existing knowledge, a proper design of heterogeneous Fenton catalyst with nanoconfinement is expected to make a fundamental change to the pollutant removal process.

The elucidation of how nanoconfinement affects the pollutant conversion pathway requires the construction of a model system with decipherable chemical environment surrounding the Fenton catalytic center. For such purpose, a metal-organic framework material UiO-66-$NH_2$-(Zr) with a well-defined chemical structure has been demonstrated to be suitable for immobilizing uniform active metal sites in various reactions[33–35]. The stable and inert UiO-66-$NH_2$-(Zr) skeleton with $Zr_6$-oxo clusters exposes abundant $\mu_3$-OH and terminal -OH/OH$_2$ groups that could anchor extraneous metal atoms (Supplementary Fig. 1). The loading of such active metal containing UiO-66-$NH_2$-(Zr) onto a proper substrate, for example graphene[24,36], would create a nanoconfined space surrounding the active metal site, generating an ideal platform for further investigation (Supplementary Fig. 2).

Herein, we describe our attempts to verify the above hypothesis by firstly anchoring Fe(III) atoms of intrinsically low Fenton reactivity on UiO-66-$NH_2$-(Zr), generating a sample referred to as UiO-66-$NH_2$-(Zr/Fe) (Fig. 1a, the views of their structures from other angles are shown in Supplementary Fig. 3). As expected, UiO-66-$NH_2$-(Zr/Fe) catalyzed Fenton reaction for the removal of phenol with a very low rate. Interestingly, when the UiO-66-$NH_2$-(Zr/Fe) nanoparticles are confined by the graphene aerogel (GA) substrate (named UiO-66-$NH_2$-(Zr/Fe)/GA, Fig. 1a), the resultant composite exhibited greatly improved reactivity for phenol removal with the rate constant two orders of magnitude higher (208 times) than the unconfined analog. The analysis of intermediates and products suggests that the GA nanoconfinement converts the carbon transfer pathways from the kinetically favored ring-opening degradation route to the thermodynamically favored oligomerization route. Our work may provide a paradigm for delicate design of high-efficiency Fenton reaction system via nanoconfinement, featuring more efficient removal of phenolic pollutants, lowered chemical dosages and carbon emission, and the generation of oligomer products for possible recovery or energy harvesting, compared to the conventional mineralization methodologies.

## Results
### Characterization of catalysts
The synthesis generated three metal-organic frameworks (MOFs) materials including UiO-66-$NH_2$-(Zr), UiO-66-$NH_2$-(Zr/Fe), and the UiO-66-$NH_2$-(Zr/Fe)/GA composite. The mass contents of Fe elements were determined by inductively coupled plasma mass spectrometry (ICP-

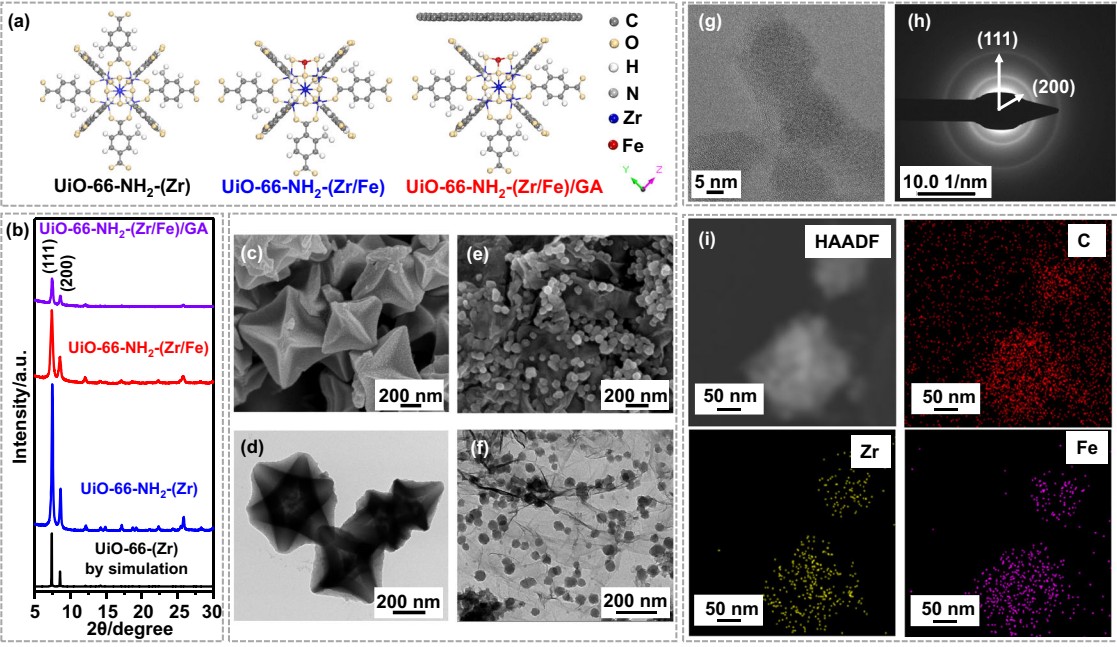

**Fig. 1 | The structural characterization of catalysts. a** Illustration of the molecular structure and (**b**) XRD patterns of UiO-66-(Zr) by simulation, UiO-66-$NH_2$-(Zr), UiO-66-$NH_2$-(Zr/Fe), and UiO-66-$NH_2$-(Zr/Fe)/GA. Representative (**c**) SEM and (**d**) TEM images of UiO-66-$NH_2$-(Zr/Fe). Representative (**e**) SEM and (**f**) TEM images of UiO-66-$NH_2$-(Zr/Fe)/GA. **g** HRTEM image, (**h**) SAED patterns, and (**i**) High-angle annular dark field (HAADF) and EDS elemental mapping images of UiO-66-$NH_2$-(Zr/Fe)/GA.

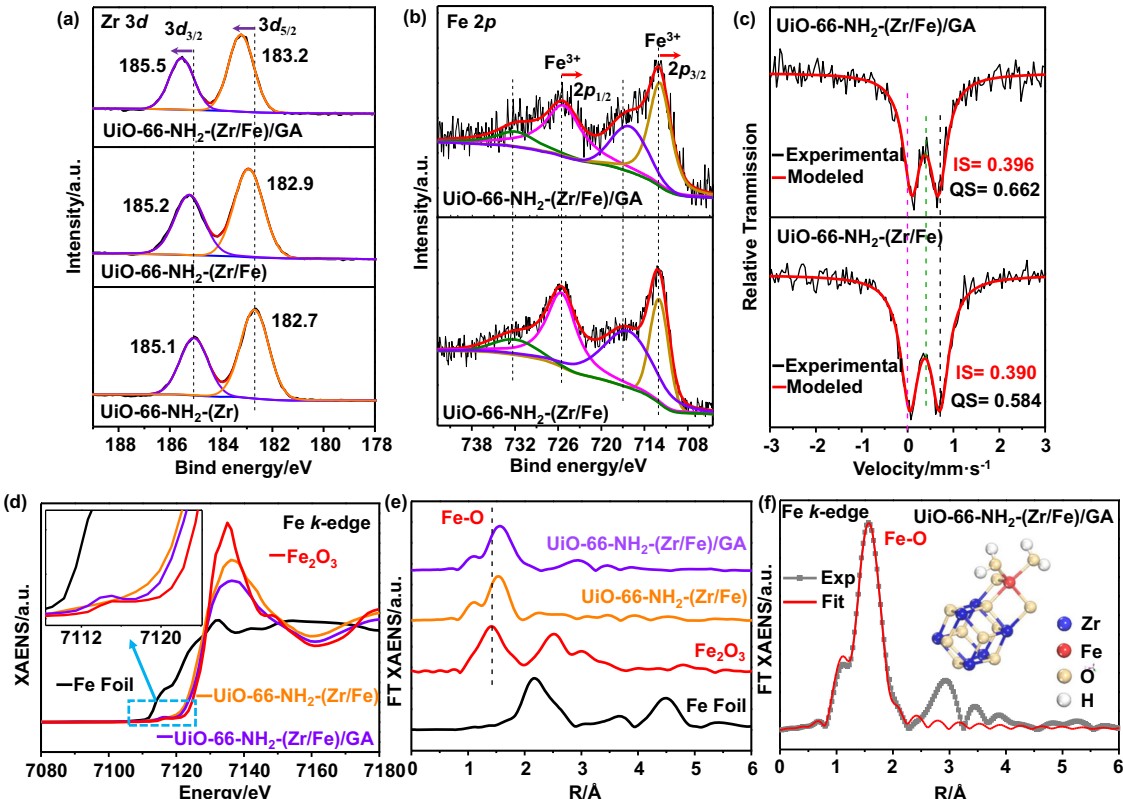

**Fig. 2 | The spectroscopic characterization of catalysts.** The deconvolutions of XPS (**a**) Zr 3*d* and (**b**) Fe 2*p* spectra. Data fitting was performed using uniform half-peak width and non-fixed peak positions. **c** $^{57}$Fe Mössbauer spectra of UiO-66-NH$_2$-(Zr/Fe) and UiO-66-NH$_2$-(Zr/Fe)/GA at 25 °C. **d** XANES Fe *k*-edge (inset: magnified spectra) and (**e**) FT-EXAFS Fe *k*-edge spectra in the *R* space of UiO-66-NH$_2$-(Zr/Fe) and UiO-66-NH$_2$-(Zr/Fe)/GA, Fe foil and Fe$_2$O$_3$ were used as references. **f** Fitting of the FT-EXAFS Fe *k*-edge curve (inset: fitting structure) of UiO-66-NH$_2$-(Zr/Fe)/GA.

MS) to be 4.4% in UiO-66-NH$_2$-(Zr/Fe) and 2.9% in UiO-66-NH$_2$-(Zr/Fe)/GA. The X-ray diffraction (XRD) patterns of the three samples are shown in Fig. 1b. UiO-66-NH$_2$-(Zr) exhibits characteristic diffraction peaks that are consistent with the results in previous studies[37,38]. For UiO-66-NH$_2$-(Zr/Fe), the presence of Fe does not change the crystalline structure of the MOFs scaffold. With respect to the UiO-66-NH$_2$-(Zr/Fe)/GA composite, the MOFs crystal structure is preserved with clear diffraction peaks of the (111) and (200) planes but with much weaker intensities.

The microscopic morphology of UiO-66-NH$_2$-(Zr/Fe) and UiO-66-NH$_2$-(Zr/Fe)/GA were examined by both scanning electron microscopy (SEM) and transmission electron microscopy (TEM). Figure 1c, d show that UiO-66-NH$_2$-(Zr/Fe) contains particles with regular octahedral shapes and sizes of several hundred nanometers to more than one micrometer. In Fig. 1e, f for UiO-66-NH$_2$-(Zr/Fe)/GA, the GA substrate could be clearly observed and the UiO-66-NH$_2$-(Zr/Fe) particles are distributed inside the GA substrate, with less regular shape but much more uniform size of ca. 40–70 nm. By comparison, GA substrate is seen to greatly reduce the size of the supported MOFs nanoparticles because the use of substrate could affect the nucleation stage and confine the growth of loaded nanoparticles[39–41]. The supported structure of UiO-66-NH$_2$-(Zr/Fe)/GA was further examined by high-resolution TEM (HRTEM) with an image shown in Fig. 1g, where the nanoparticles are observed to be sufficiently confined by the amorphous graphene. Although the crystalline lattice fringes of the UiO-66-NH$_2$-(Zr/Fe) are not distinguishable in Fig. 1g, the selected area electron diffraction (SAED) patterns in Fig. 1h clearly show the diffraction of the (111) and (200) crystal planes of the MOFs backbone. The coexistence of Zr and Fe elements in UiO-66-NH$_2$-(Zr/Fe)/GA is further depicted in the energy dispersive spectrometer (EDS) mappings in Fig. 1i. The

structure and morphology of the GA alone are illustrated in Supplementary Fig. 4.

The chemical status of metal elements in these samples were investigated by collective spectroscopic characterizations. The X-ray photoelectron spectroscopy (XPS) full spectra are presented in Supplementary Fig. 5, and the deconvolutions of the Zr 3*d* spectra are shown in Fig. 2a. In UiO-66-NH$_2$-(Zr) and UiO-66-NH$_2$-(Zr/Fe), the positions of the Zr 3*d* peaks (3*d*$_{5/2}$ and 3*d*$_{3/2}$ for Zr$^{4+}$) are very close. For UiO-66-NH$_2$-(Zr/Fe)/GA, both peaks shift slightly to higher binding energies, indicating a moderate decrease of the electron density[42]. As for the Fe 2*p* signals, Fig. 2b shows four peaks in UiO-66-NH$_2$-(Zr/Fe), in which 712.0 and 725.5 eV are attributed to the 2*p*$_{3/2}$ and 2*p*$_{1/2}$ of Fe$^{3+}$ while 716.2 and 731.9 eV are ascribed to the Fe$^{3+}$ satellite peaks[43]. These peaks shift slightly to lower binding energies by ca. 0.2–0.3 eV in UiO-66-NH$_2$-(Zr/Fe)/GA, indicating a moderate increase of the electron density which is believed to favor the Fenton reaction[44].

The $^{57}$Fe Mössbauer spectra in Fig. 2c show that the values of isomer shift (IS) are 0.390 mm·s$^{-1}$ for UiO-66-NH$_2$-(Zr/Fe) and 0.396 mm·s$^{-1}$ for UiO-66-NH$_2$-(Zr/Fe)/GA. Both values are close to 0.4 mm·s$^{-1}$, indicating the 3+ valence state of Fe in both samples[45]. A slightly larger IS value of UiO-66-NH$_2$-(Zr/Fe)/GA indicates a moderately higher electron density of Fe than that in UiO-66-NH$_2$-(Zr/Fe)[46]. The larger value of quadrupole splitting (QS) (0.662 mm·s$^{-1}$) of UiO-66-NH$_2$-(Zr/Fe)/GA suggests that the Fe environment is more distorted than that in UiO-66-NH$_2$-(Zr/Fe) (QS = 0.584 mm·s$^{-1}$)[47]. These results are consistent with those in Fig. 2b.

X-ray absorption spectroscopy (XAS) was also used to characterize UiO-66-NH$_2$-(Zr/Fe) and UiO-66-NH$_2$-(Zr/Fe)/GA. The X-ray absorption near edge structure (XANES) Fe *k*-edge spectra with Fe foil and Fe$_2$O$_3$ as references are shown in Fig. 2d, showing that the

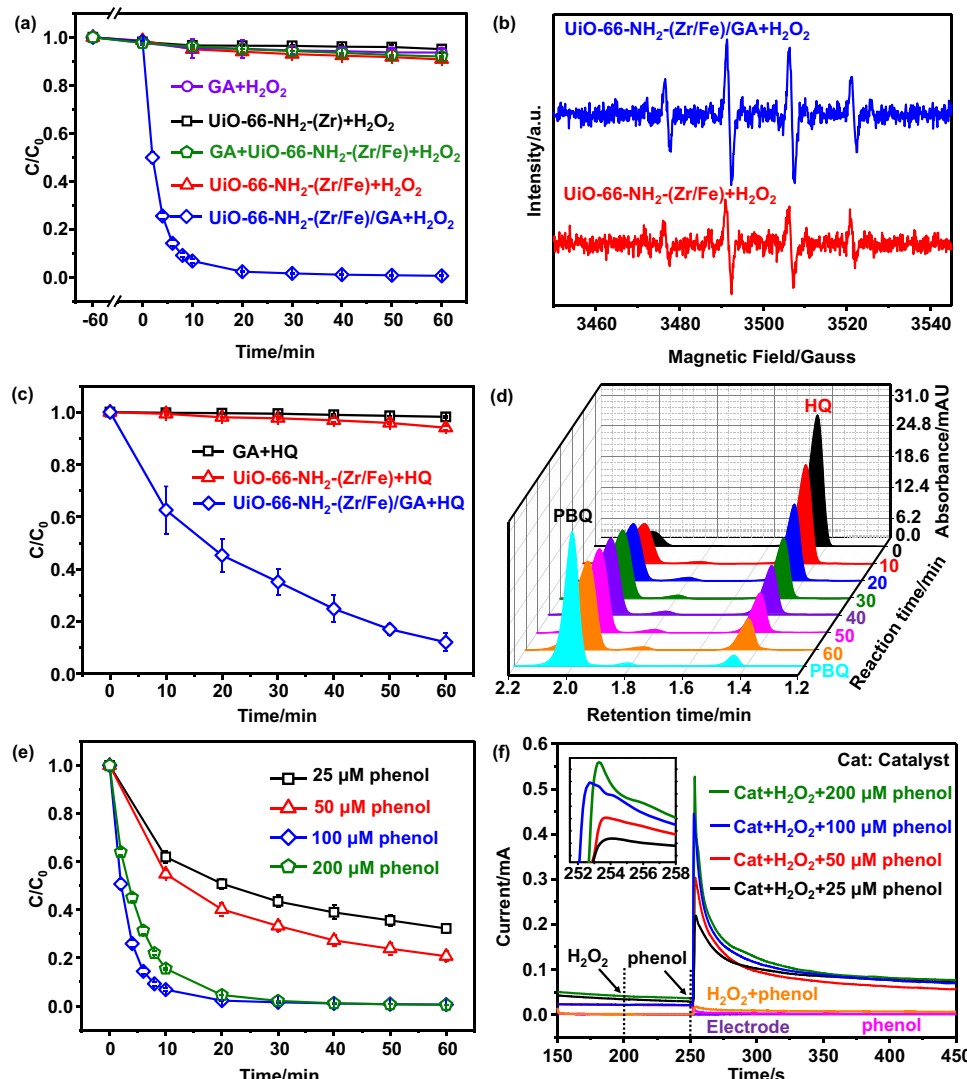

**Fig. 3 | Catalytic Fenton-like reaction for pollutant removal and mechanistic investigation. a** Removal of phenol versus time in different systems and (**b**) EPR signals by using DMPO as the trapping agent in UiO-66-NH$_2$-(Zr/Fe)+H$_2$O$_2$ and UiO-66-NH$_2$-(Zr/Fe)/GA + H$_2$O$_2$ systems at reaction time of 5 min. Conditions: initial pH = 5.0 ± 0.2, [H$_2$O$_2$] = 6 mM, [Catalyst] = 100 mg·L$^{-1}$, and [phenol] = 100 μM. **c** Removal of HQ in GA, UiO-66-NH$_2$-(Zr/Fe), and UiO-66-NH$_2$-(Zr/Fe)/GA systems

without H$_2$O$_2$. **d** HPLC chromatograms for UiO-66-NH$_2$-(Zr/Fe)/GA + HQ at different reaction time. **e** Removal of phenol by UiO-66-NH$_2$-(Zr/Fe)/GA + H$_2$O$_2$ at different phenol initial concentrations. **f** Chronoamperometry with UiO-66-NH$_2$-(Zr/Fe)/GA as the working electrode (inset: magnified spectra). Conditions: pH = 5.0 ± 0.2, voltage = +0.8 V (vs Ag/AgCl). The experiments have been carried out in triplicate, and the averaged values with standard deviations as error bars are reported.

positions of the Fe absorption edge in both samples are very close to that of Fe$_2$O$_3$ and confirming the 3+ valence state of Fe[48]. The XANES Zr *k*-edge spectra with Zr foil and ZrO$_2$ as references are shown in Supplementary Fig. 6, also confirming the 4+ valence state of Zr[49]. Moreover, the extended X-ray absorption fine structure (EXAFS) Fe *k*-edge and Zr *k*-edge signals of both samples were also recorded and shown in Supplementary Fig. 7. The corresponding Fourier transformed EXAFS (FT-EXAFS) spectra in the R space are shown in Fig. 2e (Fe *k*-edge) and Supplementary Fig. 8 (Zr *k*-edge), respectively. Figure 2e shows that there is only one Fe-O shell and no Fe-Fe shell observed in both samples, while Supplementary Fig. 8 shows that a first Zr-O shell and a second Zr-Zr shell appear in both samples. This information is crucial to indicate that the Fe atom is only located on the edge of the Zr$_6$-oxo node rather than being doped in the Zr$_6$-oxo node, because a second Fe-Fe shell would be expected if Fe participates in the formation of the main framework. The fitting of the FT-EXAFS curves reveals the coordination environment of Fe and Zr in both samples (Fig. 2f, Supplementary Figs. 9 and 10), with the best fitting parameters summarized in Supplementary Table 1. A notable message is that the coordination

number of Fe atom reduces slightly from 4.9 in UiO-66-NH$_2$-(Zr/Fe) to 4.6 in UiO-66-NH$_2$-(Zr/Fe)/GA. The above results confirm that GA gives rise to a slight modification of the electron density and coordination environment of the Fe atom on the edge of Zr$_6$-oxo node.

## Fenton-like catalytic reactions for pollutant removal

The UiO-66-NH$_2$-(Zr/Fe) and UiO-66-NH$_2$-(Zr/Fe)/GA were used as heterogeneous Fenton catalysts for the removal of phenol, a representative pollutant that has received global attention[50]. In a catalyst free system, the use of H$_2$O$_2$ only is ineffective for the phenol removal, i.e., around 4.5% removal was observed in 60 min (Supplementary Fig. 11). Figure 3a shows the removal of phenol by using UiO-66-NH$_2$-(Zr/Fe) and UiO-66-NH$_2$-(Zr/Fe)/GA as catalysts, with GA, UiO-66-NH$_2$-(Zr), and GA+UiO-66-NH$_2$-(Zr/Fe) (a physical mixture) as blank comparison. The five catalysts show negligible adsorption (<5%) for phenol in 60 min before the addition of H$_2$O$_2$. After the addition of H$_2$O$_2$, the use of GA and UiO-66-NH$_2$-(Zr) only results in 4.2% and 3.0% removal of phenol in 60 min, respectively, suggesting that both GA and UiO-66-NH$_2$-(Zr) are inert in Fenton reaction. The UiO-66-NH$_2$-(Zr/Fe) catalyst

(100 mg/L) achieves 7.9% phenol removal in 60 min, and a simple mixing of GA and UiO-66-NH$_2$-(Zr/Fe) does not change the catalytic behavior of UiO-66-NH$_2$-(Zr/Fe), i.e., 5.8% removal of phenol in 60 min for GA+UiO-66-NH$_2$-(Zr/Fe). Note that simultaneous increase of both the dosage of UiO-66-NH$_2$-(Zr/Fe) (1.0 g/L) and the reaction time (420 min) could achieve 95.4% phenol removal (Supplementary Fig. 12). These results suggest that the reactivity of UiO-66-NH$_2$-(Zr/Fe) in Fenton reaction is rather low, which is not beyond expectation because the valence state of Fe site (+3) and the pH condition (5.0) are not favored in Fenton reaction. Intriguingly, the use of UiO-66-NH$_2$-(Zr/Fe)/GA (100 mg/L) is able to remove phenol by 97.7% in only 20 min. The fitting of phenol removal kinetic data to the first-order reaction kinetics model (Supplementary Fig. 13) shows that the UiO-66-NH$_2$-(Zr/Fe)/GA + H$_2$O$_2$ system exhibits an apparent pseudo first-order rate constant ($k_{app}$) of 0.27 min$^{-1}$, which is two orders of magnitude higher (208 times) than that (0.0013 min$^{-1}$) in UiO-66-NH$_2$-(Zr/Fe)+H$_2$O$_2$.

Moreover, the removal of total organic carbon (TOC) by UiO-66-NH$_2$-(Zr/Fe)/GA + H$_2$O$_2$ is much more efficient than UiO-66-NH$_2$-(Zr/Fe)+H$_2$O$_2$ (92.2 ± 3.7% versus 3.6 ± 0.3%, Supplementary Fig. 14). With further exclusion of the possible contribution of the homogeneous Fe ions alone (Supplementary Fig. 15), homogeneous Fe ions with GA (Supplementary Fig. 16), and the effect of particle size (Supplementary Fig. 17), it is convincing that the GA nanoconfinement has resulted in a fundamental change to the catalytic reactivity/behavior of the confined UiO-66-NH$_2$-(Zr/Fe) nanoparticles, as elucidated below. After the reaction, the UiO-66-NH$_2$-(Zr/Fe)/GA catalyst retains its original crystalline structure, morphology, and chemical status (Supplementary Figs. 18–20). Furthermore, the variation of Fe electron density during the reaction is confirmed by the $^{57}$Fe Mössbauer spectra of the catalyst collected during and after the reaction (Supplementary Fig. 21).

## Mechanistic investigation during pollutant removal

We carried out analysis of the intermediates/products to investigate the phenol removal pathways in both systems. The ultrahigh performance liquid chromatograph-mass spectrometer (UHPLC-MS) spectra in Supplementary Figs. 22 and 23 show that hydroquinone (HQ) and p-benzoquinone (PBQ) are detected in both systems during reaction. Moreover, a weak signal of HQ dimer is observed in UiO-66-NH$_2$-(Zr/Fe)/GA + H$_2$O$_2$, but not in UiO-66-NH$_2$-(Zr/Fe)+H$_2$O$_2$. After the reaction, the final products in solution were characterized by gas chromatography-mass spectrometer (GC-MS). For UiO-66-NH$_2$-(Zr/Fe)+H$_2$O$_2$, the ring-opening products, i.e., acetic acid, oxalic acid, and malonic acid, are observed and summarized in Supplementary Table 2, consistent with previous studies[51]. Interestingly, for UiO-66-NH$_2$-(Zr/Fe)/GA + H$_2$O$_2$, those ring-opening products are not detected in the solution. In contrast, a mixture of oligomerized products with molecular weight ranging from around 300 to 700 are isolated from the catalyst, which was collected after the reaction using tetrahydrofuran (THF), for further matrix-assisted laser desorption/ionization time-of-flight mass spectrometry (MALDI-TOF MS) characterization (spectra are in Supplementary Fig. 24, with proposed structures listed in Supplementary Table 3). The weight balance calculation shows a yield of around 77.9% of the isolated oligomerized products based on the TOC removal, while the rest might be other products that are mineralized and/or hard to be isolated. These results confirm that phenol undergoes different degradation pathways in the two systems.

The ROS generated in both systems were then examined by using 5,5-dimethyl-1-pyrroline-1-oxide (DMPO) as a trapping agent. The electron paramagnetic resonance (EPR) spectra in Fig. 3b show that both systems exhibit a set of four peaks with intensity ratio of 1:2:2:1, which is attributed to the characteristic signals of the DMPO-•OH adduct[7]. Moreover, compared to UiO-66-NH$_2$-(Zr/Fe)+H$_2$O$_2$, the significantly stronger signals in UiO-66-NH$_2$-(Zr/Fe)/GA + H$_2$O$_2$ imply the generation of much more •OH. Results from further alcohol quenching experiments (Supplementary Fig. 25) show that excessive methanol

(MeOH) as the typical •OH quencher ($k_{MeOH, HO•}$ = 9.7 × 10$^8$ M$^{-1}$·s$^{-1}$) could effectively suppress the phenol removal in both systems, ruling out the non-radical electron transfer process[52]. With further exclusion of the contribution of other possible reactive species including singlet oxygen ($^1$O$_2$), high-valent Fe, and superoxide radicals (O$_2^{•-}$) in UiO-66-NH$_2$-(Zr/Fe)/GA + H$_2$O$_2$ (Supplementary Figs. 26–28), it is concluded that the two systems are featured with the same reactive species, i.e., •OH.

The above results indicate that after the activation of H$_2$O$_2$ to generate •OH, the reaction between phenol and •OH proceeds via different pathways in the two systems. The presence of HQ and PBQ in both systems is reasonable since •OH attacks phenol to form HQ[51]. The latter transformation of HQ to PBQ follows either Pathway I, the further attack by •OH to generate PBQ[51], or Pathway II, the sequential oxidation by Fe$^{3+}$ to form unstable semiquinone (SQ) radical and finally produce PBQ[53]. Pathway II could facilitate the regeneration of Fe$^{2+}$, resulting in a self-acceleration phenomenon that has been reported in homogeneous Fenton systems[54,55]. Such phenomenon has been scarcely reported in heterogeneous Fenton systems because HQ molecules are not prone to stay in contact with the catalyst for Pathway II. However, the dramatically improved phenol removal rate in UiO-66-NH$_2$-(Zr/Fe)/GA + H$_2$O$_2$ is a strong indication of the occurrence of the self-acceleration via Pathway II.

To verify this, we added commercial HQ molecules into three solutions containing GA, UiO-66-NH$_2$-(Zr/Fe), and UiO-66-NH$_2$-(Zr/Fe)/GA, respectively. Note that H$_2$O$_2$ is absent in these solutions. Figure 3c shows that the change of HQ concentration in GA solution is negligible (only 1.7%), suggesting a small adsorption capacity by GA. For UiO-66-NH$_2$-(Zr/Fe), there is a slight decrease of HQ concentration by 5.9% in 60 min. Remarkably, for UiO-66-NH$_2$-(Zr/Fe)/GA, the HQ concentration reduces by 87.9% in 60 min. Further high performance liquid chromatography (HPLC) results in Fig. 3d show that PBQ are gradually generated and accumulated as the consumption of HQ, indicating the transformation of HQ to PBQ. These results collectively confirm the conversion of HQ to PBQ via Pathway II, which is achieved by possibly confining HQ molecules and SQ radicals in close proximity to the anchored Fe atom on the Zr$_6$ node of UiO-66-NH$_2$-(Zr/Fe) for self-acceleration.

The self-acceleration phenomenon in UiO-66-NH$_2$-(Zr/Fe)/GA + H$_2$O$_2$ due to the occurrence of Pathway II is further exemplified in Fig. 3e, where the phenol removal efficiency increases significantly as the increase of the initial phenol concentration from 25 to 100 μM. These results (data fitting in Supplementary Fig. 29) are quite distinct from those in heterogeneous Fenton systems where the apparent pollutant removal rate, represented by $k_{app}$, should descend as the increase of the initial pollutant concentration because the steady state concentration of the reactive species, •OH, is assumed to be constant (mathematical derivation in Supplementary Note 1). In contrast, a higher initial phenol concentration in UiO-66-NH$_2$-(Zr/Fe)/GA + H$_2$O$_2$ would give rise to a higher steady state concentration of •OH, which is confirmed by the EPR results in Supplementary Fig. 30. Note that further increase of phenol concentration above 100 μM does not cause further increase in the phenol removal rate, probably because the surface Fe active site is saturated. The involvement of phenol in this self-acceleration process is also confirmed by chronoamperometry curves in Fig. 3f using UiO-66-NH$_2$-(Zr/Fe)/GA catalyst as working electrode[56]. The addition of H$_2$O$_2$ into the solution does not cause noticeable change of the current, indicating a low intrinsic reactivity of UiO-66-NH$_2$-(Zr/Fe)/GA for the activation of H$_2$O$_2$. Later, immediately after the addition of phenol into the solution, a current spike with intensity correlated to the phenol concentration is observed, confirming the participation of phenol in the electron transfer processes between the Fe(III) and the reductive intermediates in Pathway II. These results provide solid evidence for the self-acceleration via pathway II in UiO-66-NH$_2$-(Zr/Fe)/GA + H$_2$O$_2$.

## Discussion

The different reaction pathways between phenol and •OH produce disparate final products. In UiO-66-NH$_2$-(Zr/Fe)+H$_2$O$_2$, the further •OH attack proceeds via traditional ring-opening route for further mineralization. In marked contrast, in UiO-66-NH$_2$-(Zr/Fe)/GA + H$_2$O$_2$, HQ dimer is observed in solution and oligomerized products are detected in the THF eluent of the catalysts, suggesting a completely different oligomerization route. Interestingly, in a polymer synthesis scenario using Fenton reagents, HQ with a much higher concentration (50 mM) has been reported to generate oligomers, rather than being degraded[57]. Furthermore, an early study showed an activation energy of 118.5 kJ·mol$^{-1}$ for PBQ oligomerization[58], which is much smaller than that (615.3 kJ·mol$^{-1}$) for ring-opening of aromatic rings[59]. These results confirm that the oligomerization route is thermodynamically favored to the ring-opening route. In heterogeneous Fenton systems for water treatment scenarios, the concentration of phenol (100 µM) may be not high enough to allow efficient molecular collision between two or more molecules for oligomerization, which is considered to be kinetically unfavored. Intriguingly, in the GA nanoconfined environment, the intermediates are enriched with a high local concentration to eliminate the unfavored kinetic factors, triggering this thermodynamically favored oligomerization route. Different phenol removal pathways in the two systems are illustrated in Fig. 4, with the key reaction equations listed with Supplementary Fig. 31.

Furthermore, it is important to address that the consumption of the oxidant H$_2$O$_2$ in the oligomerization system is only 0.532 mM to achieve 92.2 ± 3.7% TOC removal of 0.1 mM phenol. The complete mineralization of phenol theoretically requires H$_2$O$_2$ with 14 times in molar mass[51], while a high degree of mineralization is difficult to achieve in traditional homogeneous Fe$^{2+}$+H$_2$O$_2$ systems and normally consumes much more H$_2$O$_2$ (see Supplementary Table 4 with accompanying discussion). Under similar conditions, the consumption of H$_2$O$_2$ is estimated to be reduced by more than 95% in the oligomerization system than that in the traditional homogeneous mineralization system. From the perspective of carbon emission, the oligomerization strategy allows effective reduction of carbon emission by at least 77.9% over the complete mineralization strategy for phenol removal, which accompanies the yield of the oligomerized product for possible resource recovery if value-added, or energy harvesting since it is similar to the solid sludges produced in municipal wastewater treatment plants. Consequently, this oligomerization process for phenol removal features many advantages, including faster and more efficient pollutant removal, much lower oxidant consumption, and much lower carbon emission than the existing mineralization technologies.

In summary, we have reported a proof-of-concept verification to achieve an utter alteration of carbon transfer route in the phenol removal process, i.e., from a kinetically favored ring-opening route to a thermodynamically favored oligomerization route, for an unusually improved removal efficiency in heterogeneous Fenton reaction without changing the H$_2$O$_2$ activation step. The GA nanoconfinement enables the enrichment of reductive intermediates in the close proximity to the Fe(III) atoms to facilitate the rate-limiting Fe(III) reduction step. In contrast to the traditional ring-opening route that requires excessive input of chemicals for mineralization, triggering the oligomerization route opens an avenue to develop Fenton-like technologies

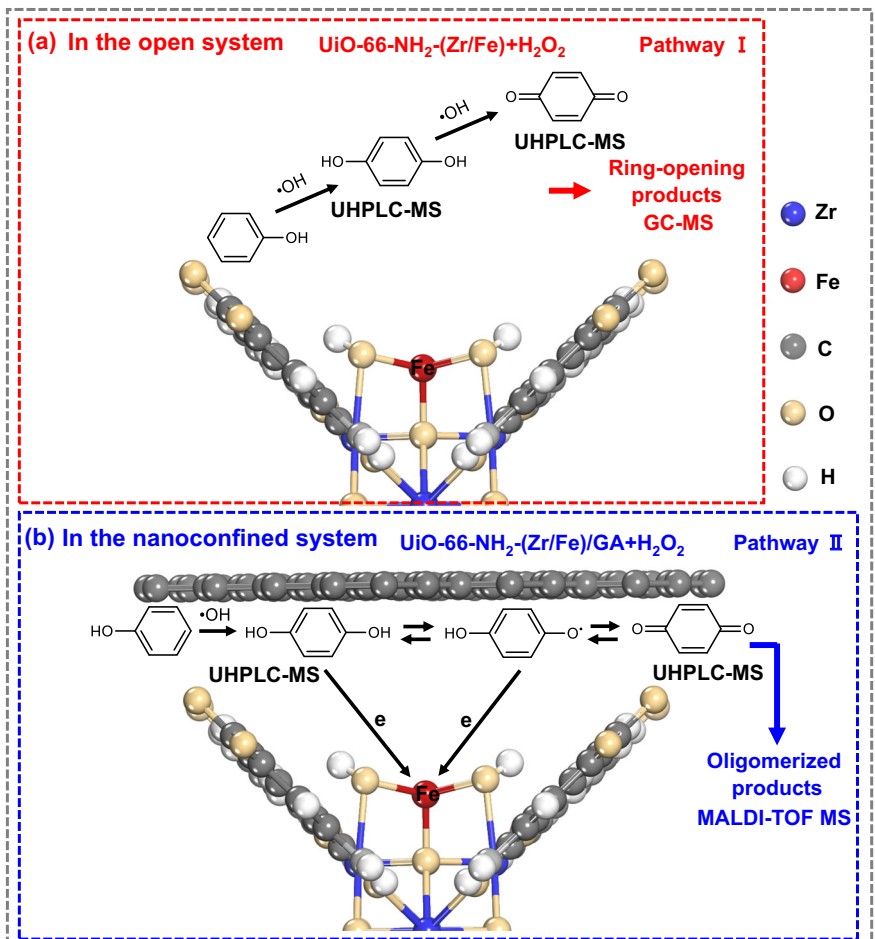

**Fig. 4 | Illustration of different phenol removal pathways. a** UiO-66-NH$_2$-(Zr/Fe)+H$_2$O$_2$ open system. **b** UiO-66-NH$_2$-(Zr/Fe)/GA + H$_2$O$_2$ nanoconfined system.

with lowered chemical consumption (comparison in Supplementary Table 5) and carbon emission, rapid and complete pollutant removal. The continuous accumulation of the oligomers could be collected after the natural precipitation on the catalyst or coagulation for further carbon recovery. Meanwhile, the catalyst exhibits excellent reusability and stability in consecutive runs after the removal of precipitated oligomer (Supplementary Fig. 32). This catalytic system is also effective for the degradation of other phenolic pollutants (Supplementary Fig. 33) and shows robust response to the interferences of complicated water matrix including the variation of solution pH (Supplementary Fig. 34), presence of various anions and natural organic substance (Supplementary Fig. 35), and temperature fluctuation (Supplementary Fig. 36). Our findings are believed to provide a direction for the design of highly efficient Fenton-like processes by regulating the carbon transfer pathways via a nanoconfinement strategy.

## Methods

### Reagents

Monolayer graphene oxide (GO) powder sample (>99 wt%, 0.5–3.0 nm in thickness) was purchased from Suzhou Tan Feng Graphene Technology Co., Ltd., China. Zirconium tetrachloride ($ZrCl_4$) was purchased from Meryer Biochemical Technology Co., Ltd, China. 2-Aminoterephthalic acid ($H_2ATA$) was purchased from Macklin Biochemical Technology Co., Ltd, China. Ferric chloride ($FeCl_3 \cdot 6H_2O$), ferrous sulfate ($FeSO_4 \cdot 7H_2O$), phenol, 2,2,6,6-tetramethylpiperidine (TEMP), methyl phenyl sulfoxide (PMSO), bisphenol A (BPA), 4-chlorophenol (4-CP), nitro blue tetrazolium (NBT), ethanol, MeOH, acetonitrile, THF, sodium chloride (NaCl), sodium sulfate ($Na_2SO_4$), sodium hydroxide (NaOH), sodium bicarbonate ($NaHCO_3$), magnesium sulfate ($MgSO_4$), humic acid (HA), hexamethyldisilazane (HMDS), and chlorotrimethylsilane (TMSCl) were purchased from Aladdin Industrial Corporation, China. Ethane diamine (EDA), N,N-dimethyl formamide (DMF), polyvinylpyrrolidone (PVP) K30, HQ, PBQ, $H_2O_2$ (30 wt%), sodium nitrate ($NaNO_3$), sodium nitrite ($NaNO_2$), hydrochloric acid (HCl), potassium acid phthalate ($C_8H_5O_4K$), potassium iodide (KI) were purchased from Sinopharm Chemical Reagent Co. Ltd., China. DMPO was purchased from Merck Life Science Technology Co., Ltd. All reagents were of analytical grade and used without further purification. All solutions in this study were prepared in ultrapure water (18.2 MΩ·cm) produced from a purification system (UPT-II-10T, Ulupure, China).

### Characterizations

The characterizations are described in Supplementary Methods.

### Preparation of Catalysts

GA was prepared through a traditional hydrothermal approach[41]. In a typical preparation, 30 mg commercial GO sample and 0.03 mL EDA were mixed in 10 mL water. Subsequently, the solution was ultrasonicated for 30 min, followed by a hydrothermal treatment at 180 °C for 5 h in a 100 mL Teflon-lined autoclave. Afterwards, a hydrogel sample was obtained after cooling and maintained in 15% ethanol for 24 h. Finally, the GA sample was acquired after freeze drying for 24 h.

The preparation of UiO-66-$NH_2$-(Zr) follows a hydrothermal method reported previously[60]. In a typical preparation: 466 mg $ZrCl_4$ and 362 mg $H_2ATA$ were mixed in a 40 mL DMF solution under constant stirring for 60 min. The mixture was then transferred to a 100 mL Teflon-lined autoclave and heated at 150 °C for 24 h. The product was washed by ethanol and water three times, followed by freeze drying for 24 h to generate UiO-66-$NH_2$-(Zr).

The preparation of UiO-66-$NH_2$-(Zr/Fe)/GA follows: 30 mg GA sample was added into a 40 mL DMF solution containing 233 mg $ZrCl_4$, 270 mg $FeCl_3 \cdot 6H_2O$, 362 mg $H_2ATA$, and 200 mg PVP under constant stirring for 60 min. The mixture was then transferred to a 100 mL Teflon-lined autoclave and heated at 150 °C for 24 h. The product was washed by ethanol and water three times, followed by freeze drying for 24 h to generate UiO-66-$NH_2$-(Zr/Fe)/GA. The UiO-66-$NH_2$-(Zr/Fe) sample was obtained using the same experimental procedures without GA. Another UiO-66-$NH_2$-(Zr/Fe)-1h sample was obtained using the heating time of only 1 h (instead of 24 h), with other procedures identical and without GA. A GA+UiO-66-$NH_2$-(Zr/Fe) sample was obtained by hand-grinding a simple mixture of GA and UiO-66-$NH_2$-(Zr/Fe) with mass ratio of 1:1.

### Examination of catalysts

All catalytic experiments in this study have been carried out in at least triplicate, and the averaged values with standard deviations as error bars are reported. All reactions were performed in a thermostatic vibrating bed (TS-100C, Shanghai Jiecheng instrument co., LTD, China), in which the temperature was kept constant at 25 °C. The initial pH of the reaction was adjusted by 0.1 M HCl or NaOH to be 5.0 ± 0.2. In a typical run, 5.0 mg catalyst was added into 50 mL phenol solution (100 μM) under magnetic stirring for 60 min to achieve adsorption equilibrium. Afterwards, 30 μL 30% $H_2O_2$ was added into the solution to trigger reaction. At predetermined time intervals, 1 mL solution was removed, filtered through a 0.22 μm glass fiber filter, and quenched with excessive ethanol for subsequent HPLC analysis. In the end of the degradation, the TOC of the solution was measured. During the cyclic experiments of phenol degradation by UiO-66-$NH_2$-(Zr/Fe)/GA + $H_2O_2$, the same catalyst was collected after reaction and washed by water three times for the use in the next cycle.

To conduct the mass balance calculation for the phenol removal by UiO-66-$NH_2$-(Zr/Fe)/GA + $H_2O_2$, the mass of all reactants and reaction volume were enlarged by eight times for a more accurate collection of final products: 40 mg UiO-66-$NH_2$-(Zr/Fe)/GA, 400 mL phenol solution (100 μM) containing 3.76 mg phenol, 240 μL 30% $H_2O_2$. The reaction as allowed to proceed for 60 min, followed by the collection of the catalyst. The catalyst was freeze dried for 24 h and then immersed by 10 mL THF under stirring for 6 h. Afterwards, the THF solution as collected and dried at 100 °C until a constant weight, in which the final oligomerized product was determined to be 2.70 mg.

## Data availability

The data generated in this study are provided within the article and the Supplementary Information file. Source data are provided with this paper.

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

## Acknowledgements

The authors thank the financial support from National Natural Science Foundation of China (22276095, J.Q.), (22236003, B.P.), (21925602, B.P.) and Natural Science Foundation of Jiangsu Province (BK20211522, J.Q.). The authors also thank Prof. Ying Guan, Prof. Xiaoyu Li, and Dr. Xiyang Wang for helpful discussions.

## Author contributions

J. Qian and B. Pan conceived the research and supervised the project; X. Zhang, J. Tang, and L. Wang performed experiments with the help of C. Wang, L. Chen, and X. Chen; X. Zhang, J. Qian, and B. Pan wrote the paper.

## Competing interests

The authors declare no competing interests.
