## [Peer Review File · Nature Communications]

Nanoconfinement-triggered oligomerization pathway for the efficient removal of phenolic pollutants via a Fenton-like reactionREVIEWER COMMENTS

Reviewer #1 (Remarks to the Author):

In this manuscript, authors have synthesized graphene aerogel (GA), which are onto the UiO-66-NH₂-(Zr) consisting Fe(III), thus the source of Fe(II) to carry out Fenton reaction. The precursor Fe(III) was created by the reduction of Fe(III) by GA. The reduces compound was oligomers of phenol, i.e., simultaneous removal of phenol. Basically, phenol is removed and reduction of Fe(III) occurred in order to have efficient Fenton reaction. Authors have fully synthesized the GA using many characterizing techniques and explanation is reasonable. Degradation of phenol was thoroughly investigated.

The main criticism of this manuscript is having a Fenton reaction system which is generating additional pollutant, oligomers. These could be separated from the water, but these have to be remediated afterwards. In other words, this research is creating efficiently other kind of secondary pollutant by removing primary pollutant. Instead of iron sludge, this system will create oligomers sludge.

Another point is that authors have to demonstrate that this newly created Fenton system is valid for only phenol or wide range of pollutants having different molecular structures.

Authors should clarify these points very clearly. Based on this the title of the manuscript may need revision.

Reviewer #2 (Remarks to the Author):

General comment: In this paper, the authors prepared a graphene aerogel (GA) supported UiO-66-NH₂-(Zr) composite with Fe(III) anchored on edge of the Zr₆-oxo cluster. The authors found that GA nanoconfinement favors the Fe(III) reduction by enriching the reductive intermediates and allows much faster phenol removal than the unconfined analog. It is believed that the current manuscript can be accepted for publication; however, some critical issues need to be addressed as follows:

1. In Abstract, some current keywords do not reflect the key points. Please revise.
2. Some properties of hydroxyl radicals such as oxidation potential, half-life, etc should be provided.
3. The format of references is inconsistent in the Introduction section (line 45 – 51). Please revise.
4. Please do not use the abbreviation of phenol as PhOH. It is not scientific.
5. Did the authors refer any reported studies to prepare catalysts? If yes, please add citation(s).
6. In Fig 1(i), it is difficult to observe the EDS mapping images for Zr and Fe. Please supply other high-resolution images.
7. More batch experiments should be investigated as competing co-existing anions and natural organic matter, temperatures, and especially pH conditions since hydroxyl radical has narrow pH active range.
8. Have the authors investigate the stability and reusability of UiO-66-NH₂-(Zr/Fe)/GA composite? It is important to examine its catalytic activity as well as stable structure after several cycle runs.

Point-to-point responses to the comments of Nature Communications

NCOMMS-23-37781

The reviewer's comments are in *italic*, our responses are in blue.

Reviewer #1

In this manuscript, authors have synthesized graphene aerogel (GA), which are onto the UiO-66-NH₂-(Zr) consisting Fe(III), thus the source of Fe(II) to carry out Fenton reaction. The precursor Fe(III) was created by the reduction of Fe(III) by GA. The reduces compound was oligomers of phenol, i.e., simultaneous removal of phenol. Basically, phenol is removed and reduction of Fe(III) occurred in order to have efficient Fenton reaction. Authors have fully synthesized the GA using many characterizing techniques and explanation is reasonable. Degradation of phenol was thoroughly investigated.

Response: We thank the reviewer for the positive comments on our manuscript, and we are particularly grateful to the Reviewer's suggestions which helped us to further improve the quality of our manuscript.

- 1. The main criticism of this manuscript is having a Fenton reaction system which is generating additional pollutant, oligomers. These could be separated from the water, but these have to be remediated afterwards. In other words, this research is creating efficiently other kind of secondary pollutant by removing primary pollutant. Instead of iron sludge, this system will create oligomers sludge.*

Response:

We thank the reviewer for this important comment. Yes, the new reaction route generates additional substance, i.e., oligomer. Here we would like to treat the formed oligomer as resource or chemical energy carrier since it is generally insoluble in water and can be readily separated after the reaction, and the separated oligomer could be possibly purified as value-added products or, if not, treated as organic sludge for energy harvesting, which has been well demonstrated in disposal of activated sludge produced in municipal wastewater treatment plants via anaerobic digestion or the drying and incineration technology. Of particular note is that, the new oligomerization route features more efficient removal of TOC from water, much less oxidant consumption and carbon emission, as we have demonstrated in the paper. Consequently, we believe the alternation of carbon transfer route gives a greener alternative to traditional mineralization in water remediation. We envision future studies that could make an appropriate use of these oligomer products.

For clarification, we have added the corresponding contents in the revised manuscript: "..., demonstrate a greener alternative for the removal of organic pollutants, featuring more efficient pollutant removal, lowered chemical dosages and carbon emission, and the generation of oligomer products for possible recovery or energy harvesting.", "From the perspective of carbon emission, the oligomerization strategy allows effective

reduction of carbon emission by at least 77.9 % over the complete mineralization strategy, which accompanies the yield of the oligomerized product for possible resource recovery if value-added, or energy harvesting since it is similar to the solid sludges produced in municipal wastewater treatment plants.”

2. *Another point is that authors have to demonstrate that this newly created Fenton system is valid for only phenol or wide range of pollutants having different molecular structures.*

Response: We thank the reviewer for this important comment, which helps us to demonstrate the versatility of our system. In Fig. S33, we have already shown the removal of bisphenol A (BPA) and 4-chlorophenol (4-CP) by the UiO-66-NH₂-(Zr/Fe)/GA+H₂O₂ nanoconfined system. Similar removal behavior to that of phenol, i.e., accelerated removal as the increase of the pollutant initial concentration, is observed.

To order to further address this comment, we carried out additional experiments to examine the removal of other model pollutants of interest, including rhodamine B (RhB), sulfamethoxazole (SMX), carbamazepine (CBZ), 4-hydroxybenzoic acid (4-HBA), and atrazine (ATZ). The results are shown in Fig. R1. Fig. R1a shows that the UiO-66-NH₂-(Zr/Fe)/GA+H₂O₂ nanoconfined system could effectively remove 87.1 % RhB, 71.7 % SMX, 80.4 % CBZ, and 85.4% 4-HBA, in 60 min. Note that the system is ineffective for the removal of ATZ, i.e., only 14.1 % in 60 min. Interestingly, Fig. R1b shows that when the ATZ is mixed with phenol, the removal of ATZ increases from 20.2 % (ATZ alone) to 54.0 % (in ATZ/phenol mixture), implying that the self-acceleration of phenol removal in the nanoconfined system could also enhance the removal of other refractory organic pollutants. These results are presented as Fig. S34 in revised SI, with accompanying discussion.

Fig. R1. (a) Degradation of various pollutants by the UiO-66-NH₂-(Zr/Fe)/GA+H₂O₂ system, RhB: rhodamine B, SMX: sulfamethoxazole, CBZ: carbamazepine, 4-HBA: 4-hydroxybenzoic acid, ATZ: atrazine. (b) The degradation of ATZ, phenol, and ATZ/phenol mixture by UiO-66-NH₂-(Zr/Fe)/GA+H₂O₂. Conditions: pH = 5.0 ± 0.2, [H₂O₂] = 6 mM, [Catalyst] = 100 mg·L⁻¹, [Pollutants] = 100 μM for (a), [Phenol] = 100 μM and [ATZ] = 25 μM for (b).

3. *Authors should clarify these points very clearly. Based on this the title of the manuscript may need revision.*

Response: We thank you for this suggestion. Based on the above comments, we have revised the title to be: “Nanoconfinement triggers a greener Fenton-like reaction via oligomerization”.

Reviewer #2

In this paper, the authors prepared a graphene aerogel (GA) supported UiO-66-NH₂-(Zr) composite with Fe(III) anchored on edge of the Zr₆-oxo cluster. The authors found that GA nanoconfinement favors the Fe(III) reduction by enriching the reductive intermediates and allows much faster phenol removal than the unconfined analog. It is believed that the current manuscript can be accepted for publication; however, some critical issues need to be addressed as follows.

Response: We really appreciate such positive evaluation, and we are especially grateful to the comments which helped us to improve the quality of our paper for publication in Nature Communications.

1. *In Abstract, some current keywords do not reflect the key points. Please revise.*

Response: Thank you for this suggestion. We have changed the keywords to “Fenton reaction, mineralization, nanoconfinement, oligomerization, green chemistry”

2. *Some properties of hydroxyl radicals such as oxidation potential, half-life, etc should be provided.*

Response: Thank you for this suggestion. We have added this information into the revised manuscript “(oxidation potential: 2.80 V, half-life: < 5 μs).”

3. *The format of references is inconsistent in the Introduction section (line 45-51). Please revise.*

Response: Thank you for this suggestion. We have made the format of the references consistent in the revised manuscript.

4. *Please do not use the abbreviation of phenol as PhOH. It is not scientific.*

Response: Thank you for this suggestion. We have changed “PhOH” to “phenol” throughout the revised manuscript and the SI.

5. *Did the authors refer any reported studies to prepare catalysts? If yes, please add citation(s).*

Response: Thank you for this suggestion. For the preparation of GA and UiO-66-NH₂-(Zr), we have added the reference [41] and [60], respectively, to the revised manuscript. Note that our work reports the first example of UiO-66-NH₂-(Zr/Fe)/GA composite as catalyst.

6. *In Fig 1(i), it is difficult to observe the EDS mapping images for Zr and Fe. Please supply other high-resolution images.*

Response: Thank you for this suggestion. In order to obtain EDS mappings with better quality, we have carried out additional EDS elemental mapping characterizations with higher magnifications on the UiO-66-NH₂-(Zr/Fe)/GA sample. The new figures are shown in Fig. R2, and used to replace the original Fig. 1i in the revised manuscript.

Fig. R2. High-Angle Annular Dark Field (HAADF) Energy Dispersive Spectrometer (EDS) elemental mapping images of UiO-66-NH₂-(Zr/Fe)/GA.

7. *More batch experiments should be investigated as competing co-existing anions and natural organic matter, temperatures, and especially pH conditions since hydroxyl radical has narrow pH active range.*

Response: Thank you for this suggestion. We have carried out additional experiments to examine the effects of co-existing anions and natural organic matter (Fig. R3a), temperatures (Fig. R3b and c), and pH conditions (Fig. R3d) for phenol degradation in this system. The results have been added to the revised SI as Fig. S35 to S37 with accompanying discussion.

Fig. R3a shows that HCO₃⁻ does not cause significant effects on the phenol removal, while Cl⁻, SO₄²⁻, and NO₃⁻ inhibits the phenol removal to different extents. These results are consistent with other systems reported previously [*Chem. Eng. J.* **2021**, 411, 128392; *Chem. Eng. J.* **2021**, 414, 128669], presumably due to the reaction between •OH and these anions. Moreover, the presence of natural organic substance HA greatly suppresses the phenol removal because of the competitive consumption of the •OH reactive species by HA [*Environ. Sci. Technol.* **2022**, 56, 11111-11131].

Fig. R3b shows that the phenol removal by the UiO-66-NH₂-(Zr/Fe)/GA+H₂O₂ system is affected by the reaction temperature. Based on the first-order reaction rate constant at different temperatures, we could build the Arrhenius plot in Fig. R3c, from which the activation energy of the reaction is calculated to be 58.2 kJ/mol.

Fig. R3d shows that the UiO-66-NH₂-(Zr/Fe)/GA+H₂O₂ system exhibits excellent phenol removal at initial pH of 4.0 and 5.0. When the initial pH is further increased to 5.5, the removal efficiency decreases obviously, 63.2% in 20 min and 81.9% in 60 min. The catalytic system is almost unable to remove phenol when the initial pH is increased to 6.0 and 7.0. The reduction of removal efficiency is apparently due to the reduced catalytic reactivity of Fe atoms in heterogeneous Fenton reactions under elevated pH conditions and reduced activity of •OH [*Environ. Sci. Technol.* **2018**, 52, 7043-7053; *Environ. Sci. Technol.* **2018**, 52, 5367-5377; *Nano Res.* **2021**, 14, 2383-2389].

Fig. R3. (a) Removal of phenol by UiO-66-NH₂-(Zr/Fe)/GA/H₂O₂ (blank) in the presence of various coexisting ions and natural organic substance, i.e., NO₃⁻ (10 mM), SO₄²⁻ (10 mM), Cl⁻ (10 mM), HCO₃⁻ (10 mM), and HA (10 mg/L). (b) Removal of phenol by UiO-66-NH₂-(Zr/Fe)/GA/H₂O₂ at different temperatures. (c) The Arrhenius curve of phenol degradation at different temperatures. (d) Removal of phenol by the UiO-66-NH₂-(Zr/Fe)/GA+H₂O₂ system at different initial pH. Conditions: initial pH = 5.0 ± 0.2, [H₂O₂] = 6 mM, [Catalyst] = 100 mg·L⁻¹, and [Phenol] = 100 μM.

8. *Have the authors investigate the stability and reusability of UiO-66-NH₂-(Zr/Fe)/GA composite? It is important to examine its catalytic activity as well as stable structure after several cycle runs.*

Response: Thank you for this comment. Yes, we have carried out XRD, XPS and Mössbauer characterizations to investigate the stability and reusability of the used UiO-66-NH₂-(Zr/Fe)/GA catalyst. Results in Fig. S18, S20, and S21 confirm that the catalyst retains its original crystalline structure and chemical status after the catalytic reaction. To further demonstrate the morphology of the used sample, we have carried out additional HRTEM characterizations on the used UiO-66-NH₂-(Zr/Fe)/GA catalyst, as shown in Fig. R4. It is seen that the morphology of the catalyst remains unchanged after the reaction. These new results are added as Fig. S19 in the revised SI.

Fig. R4. (a) TEM image, (b) and (c) HRTEM images, (d) Selected area electron diffraction (SAED) patterns, and (e-h) EDS elemental mapping of the used UiO-66-NH₂-(Zr/Fe)/GA sample.

In terms of the catalytic activity, results in Fig. S32 with accompanying discussion show that the UiO-66-NH₂-(Zr/Fe)/GA catalyst demonstrate excellent reusability and stability in consecutive runs.

REVIEWERS' COMMENTS

Reviewer #1 (Remarks to the Author):

Further comments are made in **red text**. **The manuscript needs further revision.**

Reviewer #1

In this manuscript, authors have synthesized graphene aerogel (GA), which are onto the UiO-66-NH₂(Zr) consisting Fe(III), thus the source of Fe(II) to carry out Fenton reaction. The precursor Fe(III) was created by the reduction of Fe(III) by GA. The reduces compound was oligomers of phenol, i.e., simultaneous removal of phenol. Basically, phenol is removed and reduction of Fe(III) occurred in order to have efficient Fenton reaction. Authors have fully synthesized the GA using many characterizing techniques and explanation is reasonable. Degradation of phenol was thoroughly investigated.

Response: We thank the reviewer for the positive comments on our manuscript, and we are particularly grateful to the Reviewer's suggestions which helped us to further improve the quality of our manuscript.

Re-Review: Argument made here need further clarification as this reviewer is not convinced about secondary sludge. Since the approach stated here generate secondary sludge how authors justify added "greener" in the title.

- 1. The main criticism of this manuscript is having a Fenton reaction system which is generating additional pollutant, oligomers. These could be separated from the water, but these have to be remediated afterwards. In other words, this research is creating efficiently other kind of secondary pollutant by removing primary pollutant. Instead of iron sludge, this system will create oligomers sludge.*

Response:

We thank the reviewer for this important comment. Yes, the new reaction route generates additional substance, i.e., oligomer. Here we would like to treat the formed oligomer as resource or chemical energy carrier since it is generally insoluble in water and can be readily separated after the reaction, and the separated oligomer could be possibly purified as value-added products or, if not, treated as organic sludge for energy harvesting, which has been well demonstrated in disposal of activated sludge produced in municipal wastewater treatment plants via anaerobic digestion or the drying and incineration technology. Of particular note is that, the new oligomerization route features more efficient removal of TOC from water, much less oxidant consumption and carbon emission, as we have demonstrated in the paper. Consequently, we believe the alternation of carbon transfer route gives a greener alternative to traditional mineralization in water remediation. We envision future studies that could make an appropriate use of these oligomer products.

Re-Reviewer: It seems authors rely on oligomerization for the removal process. This suggests that the selective generation of radicals that combine to yield oligomers. This may be only limited to phenol and hence how authors could convince the readers that this is universally applied to other kinds of pollutants besides phenols. As shown

below other pollutants could not be removed through oligomerization mechanism. The stated 80% removal of TOC confines to phenols.

For clarification, we have added the corresponding contents in the revised manuscript: "..., demonstrate a greener alternative for the removal of organic pollutants, featuring more efficient pollutant removal, lowered chemical dosages and carbon emission, and the generation of oligomer products for possible recovery or energy harvesting.", "From the perspective of carbon emission, the oligomerization strategy allows effective reduction of carbon emission by at least 77.9 % over the complete mineralization strategy, which accompanies the yield of the oligomerized product for possible resource recovery if value-added, or energy harvesting since it is similar to the solid sludges produced in municipal wastewater treatment plants."

2. *Another point is that authors have to demonstrate that this newly created Fenton system is valid for only phenol or wide range of pollutants having different molecular structures.*

Response: We thank the reviewer for this important comment, which helps us to demonstrate the versatility of our system. In Fig. S33, we have already shown the removal of bisphenol A (BPA) and 4-chlorophenol (4-CP) by the UiO-66-NH₂-(Zr/Fe)/GA+H₂O₂ nanoconfined system. Similar removal behavior to that of phenol, i.e., accelerated removal as the increase of the pollutant initial concentration, is observed.

To order to further address this comment, we carried out additional experiments to examine the removal of other model pollutants of interest, including rhodamine B (RhB), sulfamethoxazole (SMX), carbamazepine (CBZ), 4-hydroxybenzoic acid (4-HBA), and atrazine (ATZ). The results are shown in Fig. R1. Fig. R1a shows that the UiO-66-NH₂-(Zr/Fe)/GA+H₂O₂ nanoconfined system could effectively remove 87.1 % RhB, 71.7 % SMX, 80.4 % CBZ, and 85.4% 4-HBA, in 60 min. Note that the system is ineffective for the removal of ATZ, i.e., only 14.1 % in 60 min. Interestingly, Fig. R1b shows that when the ATZ is mixed with phenol, the removal of ATZ increases from 20.2 % (ATZ alone) to 54.0 % (in ATZ/phenol mixture), implying that the self-acceleration of phenol removal in the nanoconfined system could also enhance the removal of other refractory organic pollutants. These results are presented as Fig. S34 in revised SI, with accompanying discussion.

Re-Reviewer: This again suggests that the proposed system confines to phenols and not a system that can be applied for a wide range of pollutants. Authors have not showed that the removal enhancement of ATZ resulted due to oligomerization. ATZ/phenol may take another route rather than oligomerization and since authors monitor only ATZ and pushed conclusions to preconceived assessment with no experimental evidence.

Fig. R1. (a) Degradation of various pollutants by the UiO-66-NH₂-(Zr/Fe)/GA+H₂O₂ system, RhB: rhodamine B, SMX: sulfamethoxazole, CBZ: carbamazepine, 4-HBA: 4-hydroxybenzoic acid, ATZ: atrazine. (b) The degradation of ATZ, phenol, and ATZ/phenol mixture by UiO-66-NH₂-(Zr/Fe)/GA+H₂O₂. Conditions: pH = 5.0 ± 0.2, [H₂O₂] = 6 mM, [Catalyst] = 100 mg·L⁻¹, [Pollutants] = 100 μM for (a), [Phenol] = 100 μM and [ATZ] = 25 μM for (b).

3. Authors should clarify these points very clearly. Based on this the title of the manuscript may need revision.

Response: We thank you for this suggestion. Based on the above comments, we have revised the title to be: "Nanoconfinement triggers a greener Fenton-like reaction via oligomerization".

Re-Reviewer: Authors need to make stronger arguments or a better case that oligomers can be separated easily and then converted to usable products. What are the solubility of oligomers. If it is easy to separate oligomers, such separation can be shown in this manuscript.

Overall, this manuscript needs further clarification with stronger scientific writing with less speculation to avoid that the proposed system does not produce secondary sludge.

REVIEWER COMMENTS

Reviewer #2 (Remarks to the Author):

The authors have addressed the concerns raised in the original manuscript and now it can be considered for publication.

Point-to-point responses to the comments of Nature Communications

NCOMMS-23-37781

The reviewer's comments are in *italic*, our responses are in blue.

Further comments are made in red text.

Our further responses to these new comments are in green.

Reviewer #1

In this manuscript, authors have synthesized graphene aerogel (GA), which are onto the UiO-66-NH₂-(Zr) consisting Fe(III), thus the source of Fe(II) to carry out Fenton reaction. The precursor Fe(III) was created by the reduction of Fe(III) by GA. The reduces compound was oligomers of phenol, i.e., simultaneous removal of phenol. Basically, phenol is removed and reduction of Fe(III) occurred in order to have efficient Fenton reaction. Authors have fully synthesized the GA using many characterizing techniques and explanation is reasonable. Degradation of phenol was thoroughly investigated.

Response: We thank the reviewer for the positive comments on our manuscript, and we are particularly grateful to the Reviewer's suggestions which helped us to further improve the quality of our manuscript.

Re-Review: Argument made here need further clarification as this reviewer is not convinced about secondary sludge. Since the approach stated here generate secondary sludge how authors justify added "greener" in the title.

Response: Again, we thank the reviewer for this suggestive comment. We understand that the reviewer is concerned about the "secondary sludge", and especially the "greener" concept in the title. To address this comment, we now revised the title to be more neutral, specific, and clear, i.e., "*Nanoconfinement triggers the oligomerization pathway for efficient removal of phenolic pollutants in Fenton-like reaction*". As one could see, we remove the "greener" or "sustainable" from the title, and moreover, we restrict our findings to "phenolic pollutants", which is also emphasized by the reviewer.

- 1. The main criticism of this manuscript is having a Fenton reaction system which is generating additional pollutant, oligomers. These could be separated from the water, but these have to be remediated afterwards. In other words, this research is creating efficiently other kind of secondary pollutant by removing primary pollutant. Instead of iron sludge, this system will create oligomers sludge.*

Response: We thank the reviewer for this important comment. Yes, the new reaction route generates additional substance, i.e., oligomer. Here we would like to treat the formed oligomer as resource or chemical energy carrier since it is generally insoluble in water and can be readily separated after the reaction, and the separated oligomer could be possibly purified as value-added products or, if not, treated as organic sludge for energy harvesting, which has been well demonstrated in disposal of activated sludge

produced in municipal wastewater treatment plants via anaerobic digestion or the drying and incineration technology. Of particular note is that, the new oligomerization route features more efficient removal of TOC from water, much less oxidant consumption and carbon emission, as we have demonstrated in the paper. Consequently, we believe the alternation of carbon transfer route gives a greener alternative to traditional mineralization in water remediation. We envision future studies that could make an appropriate use of these oligomer products.

Re-Reviewer: It seems authors rely on oligomerization for the removal process. This suggests that the selective generation of radicals that combine to yield oligomers. This may be only limited to phenol and hence how authors could convince the readers that this is universally applied to other kinds of pollutants besides phenols. As shown below other pollutants could not be removed through oligomerization mechanism. The stated 80% removal of TOC confines to phenols.

Response: We agree with the reviewer's comments. We have changed the title to be "Nanoconfinement triggers the oligomerization pathway for efficient removal of phenolic pollutants in Fenton-like reaction", which limits our findings to phenolic pollutants that are well demonstrated in our study. We have also made clear that the removal of TOC and lowered carbon emission are confined to phenols, as illustrated by blue text in the revised version. We also replaced the original keyword "green chemistry" by "phenolic pollutants". Moreover, since we have made restriction to "phenolic pollutants" in the title, we removed the results on the degradation of other pollutants (original Figure S34 in SI) from the revised supporting formation.

For clarification, we have added the corresponding contents in the revised manuscript: "..., demonstrate a greener alternative for the removal of organic pollutants, featuring more efficient pollutant removal, lowered chemical dosages and carbon emission, and the generation of oligomer products for possible recovery or energy harvesting.", "From the perspective of carbon emission, the oligomerization strategy allows effective reduction of carbon emission by at least 77.9 % over the complete mineralization strategy, which accompanies the yield of the oligomerized product for possible resource recovery if value-added, or energy harvesting since it is similar to the solid sludges produced in municipal wastewater treatment plants."

Response: Based on the reviewer's comment, we have removed "greener" in the final version. Other descriptions are valid based on our results, with restriction to phenols. The text now reads as "Our work may provide a new paradigm for delicate design of high-efficiency Fenton reaction system via nanoconfinement, featuring more efficient removal of phenolic pollutants,..." "From the perspective of carbon emission, the oligomerization strategy allows effective reduction of carbon emission by at least 77.9 % over the complete mineralization strategy for phenol removal, ..."

2. *Another point is that authors have to demonstrate that this newly created Fenton system is valid for only phenol or wide range of pollutants having different molecular structures.*

Response: We thank the reviewer for this important comment, which helps us to demonstrate the versatility of our system. In Fig. S33, we have already shown the removal of bisphenol A (BPA) and 4-chlorophenol (4-CP) by the UiO-66-NH₂-(Zr/Fe)/GA+H₂O₂ nanoconfined system. Similar removal behavior to that of phenol, i.e., accelerated removal as the increase of the initial pollutant concentration, is observed.

To order to further address this comment, we carried out additional experiments to examine the removal of other model pollutants of interest, including rhodamine B (RhB), sulfamethoxazole (SMX), carbamazepine (CBZ), 4-hydroxybenzoic acid (4-HBA), and atrazine (ATZ). The results are shown in Fig. R1. Fig. R1a shows that the UiO-66-NH₂-(Zr/Fe)/GA+H₂O₂ nanoconfined system could effectively remove 87.1 % RhB, 71.7 % SMX, 80.4 % CBZ, and 85.4% 4-HBA, in 60 min. Note that the system is ineffective for the removal of ATZ, i.e., only 14.1 % in 60 min. Interestingly, Fig. R1b shows that when the ATZ is mixed with phenol, the removal of ATZ increases from 20.2 % (ATZ alone) to 54.0 % (in ATZ/phenol mixture), implying that the self-acceleration of phenol removal in the nanoconfined system could also enhance the removal of other refractory organic pollutants. These results are presented as Fig. S34 in revised SI, with accompanying discussion.

Re-Reviewer: This again suggests that the proposed system confines to phenols and not a system that can be applied for a wide range of pollutants. Authors have not showed that the removal enhancement of ATZ resulted due to oligomerization. ATZ/phenol may take another route rather than oligomerization and since authors monitor only ATZ and pushed conclusions to preconceived assessment with no experimental evidence.

Fig. R1. (a) Degradation of various pollutants by the UiO-66-NH₂-(Zr/Fe)/GA+H₂O₂ system, RhB: rhodamine B, SMX: sulfamethoxazole, CBZ: carbamazepine, 4-HBA: 4-hydroxybenzoic acid, ATZ: atrazine. (b) The degradation of ATZ, phenol, and ATZ/phenol mixture by UiO-66-NH₂-(Zr/Fe)/GA+H₂O₂. Conditions: pH = 5.0 ± 0.2, [H₂O₂] = 6 mM, [Catalyst] = 100 mg·L⁻¹, [Pollutants] = 100 μM for (a), [Phenol] = 100 μM and [ATZ] = 25 μM for (b).

Response: Thank you for this comment. Based on the reviewer's comment, we have restricted our title to phenolic pollutants. Consequently, we think the results in Figure R1 (Figure S34 in SI) are no longer necessary.

3. *Authors should clarify these points very clearly. Based on this the title of the manuscript may need revision.*

Response: We thank you for this suggestion. Based on the above comments, we have revised the title to be: “Nanoconfinement triggers a greener Fenton-like reaction via oligomerization”.

Re-Reviewer: Authors need to make stronger arguments or a better case that oligomers can be separated easily and then converted to usable products. What are the solubility of oligomers. If it is easy to separate oligomers, such separation can be shown in this manuscript.

Response: Yes, we have changed the title to be “*Nanoconfinement triggers the oligomerization pathway for efficient removal of phenolic pollutants in Fenton-like reaction*”. We have also decided to remove the “easily” from the text.

With respect to the solubility of oligomers, it is reasonable to suggest that the solubility is very low as these oligomers were not detected by HPLC-MS (only HQ dimer was observed), as shown in Figure S23 and described in the main text “a weak signal of HQ dimer is observed”. Our results have shown that the separation of oligomer from the solution was easy, but the separation of oligomer from the used catalyst requires solvent washing (Figure S24 and S32 in SI). Since the reviewer is still doubtful of “easy separation”, we have removed such description.

With respect to the “conversion to usable product”, we have made sure to claim “... future possible resource recovery...”. It is clear that the further resource recovery on the oligomer is beyond the scope of the current study, as clearly defined by the new title.

Overall, this manuscript needs further clarification with stronger scientific writing with less speculation to avoid that the proposed system does not produce secondary sludge.

Response: Thank you again for your comments. As you could see, we have made significant changes to the manuscript to avoid the possible speculations. All the descriptions in the revised version are based on our experimental evidences.

REVIEWERS' COMMENTS

Reviewer #1 (Remarks to the Author):

This reviewer is satisfied with responses and have no further comments.